# Decreased Expression of Leptin among Patients with Shoulder Stiffness

**DOI:** 10.3390/life12101588

**Published:** 2022-10-12

**Authors:** Jih-Yang Ko, Sung-Hsiung Chen, Re-Wen Wu, Kuan-Ting Wu, Chieh-Cheng Hsu, Shu-Jui Kuo

**Affiliations:** 1Department of Orthopedic Surgery, Kaohsiung Chang Gung Memorial Hospital, Kaohsiung 833401, Taiwan; 2Center for Shockwave Medicine and Tissue Engineering, Department of Medical Research, Kaohsiung Chang Gung Memorial Hospital, Kaohsiung 833401, Taiwan; 3School of Medicine, China Medical University, Taichung 404328, Taiwan; 4Department of Orthopedic Surgery, China Medical University Hospital, Taichung 404327, Taiwan

**Keywords:** shoulder stiffness, leptin, interleukin-6, interleukin-10, interleutkin-13, interleukin-1β

## Abstract

Shoulder stiffness (SS) is a disease that is fibroblastic and inflammatory in nature. Leptin is an adipokine-mediating the fibroblastic and inflammatory processes of various diseases. Our study tried to investigate the role of leptin in SS pathogenesis. Subacromial bursa from stiff and non-stiff shoulders were obtained for reverse transcription-polymerase chain reaction (RT-PCR) analysis and immunoblotting. Subacromial fluid was obtained for enzyme-linked immunosorbent assay. We showed that the expression level of leptin was lower in the subacromial bursae from the stiff shoulders in RT-PCR analysis (*p* < 0.001) and immunoblotting (*p* < 0.001). The concentration of leptin was also lower in the subacromial fluid derived from stiff shoulders. The leptin level in the subacromial fluid was positively associated with the constant score, total range of motion, flexion, abduction, and external rotation. The synovial fibroblasts derived from stiff shoulder-retrieved subacromial bursa were treated by 0, 1, and 3 μM leptin. Under RT-qPCR analysis, leptin was shown to dose-dependently decrease the transcription of IL-6, IL-10, and IL-13, but without impact on IL-1β and IL-4 (*p* < 0.001, *p* = 0.001, *p* = 0.001, *p* = 0.137, and *p* = 0.883 by ANOVA test, respectively). These results shed light on the role of leptin in orchestrating the disease processes of SS.

## 1. Introduction

Shoulder stiffness (SS) is a prevalent but not completely understood disease manifested by pain, a decreased range of motion (ROM) (especially external rotation), and functional incapacity. The lifetime SS prevalence is about 2 to 5% of the general population [1]. SS is most common in the fifth and sixth decades of life, with the peak age in the mid-50s. Women are more often affected than men [2]. Some patients suffer from SS associated with rotator cuff lesions, and some patients are cryptogenic [3]. Although agreement on the pathogenesis of SS with rotator cuff lesions is not unified, it seems that it is fibroblastic and inflammatory in nature [4]. Some recognized risk factors for SS development are diabetes, Dupuytren’s syndrome, cancer, shoulder trauma, smoking, heart and neck operation, or chronic regional pain syndrome [5]. Many risk factors mentioned above harbor fibroblastic and inflammatory characteristics.

While SS was often considered a self-limiting and harmless disease, patients with substantial SS suffer from long-term disabilities, with some patients reporting discomfort 6 years after SS onset, posing a substantial burden to the healthcare system [6]. The economic impact of SS is underscored by its predilection for adults of working age (8.2% for men and 10.1% for women) [7]. Overall costs per case for cases with posttraumatic and postoperative FS were unignorably CHF 34,000 per case [8].

Various treatment modalities, including rehabilitation, medication, extracorporeal shockwave therapy, physiotherapy protocols, and surgery, are available; however, the present literature research does not support a definitively effective therapy [6,9,10]. The lack of a definitely effective therapy underscores the importance of deepening the understanding of the pathogenesis of SS.

Leptin, a small (16 kDa) nonglycosylated peptide hormone encoded by the obese (ob) gene, is synthesized mainly in white adipose tissue [11,12]. Leptin serves as an anorexic peptide that is primarily acknowledged for its role as a hypothalamic mediator of food intake, body weight, and fat stores [13]. Growing evidence also suggests that, in addition to the regulation of energy homeostasis, leptin also plays a key role in glucose metabolism [14]. Serum leptin level is also shown to be lower in non-obese subjects with type 2 diabetes [15].

In addition to the endocrinologic problems, leptin also affects the disease processes of various musculoskeletal diseases, such as knee osteoarthritis [16]. Leptin has been shown to orchestrate the fibroblastic and inflammatory processes of various human diseases. It is thus intriguing whether leptin could be involved in the disease processes of fibroblastic and inflammatory musculoskeletal disease, such as SS.

Interestingly, SS seems to be the musculoskeletal disease with prominent susceptibility among diabetic patients. Diabetic patients are more likely to develop SS and more likely to require operative management [17]. Diabetes seems to be the shared factor between leptin and SS.

Despite the fact that SS is known to be fibroblastic and inflammatory in nature, and leptin could orchestrate the fibroblastic and inflammatory processes in various human diseases as well as affect the disease processes of musculoskeletal diseases, the role of leptin in the disease processes of SS is unknown at present. The objective of our study was to investigate the role of leptin in the disease processes of SS and its association with clinical parameters. We hypothesize that leptin could be involved in the pathogenesis of SS, and the expression level of leptin could be associated with clinical parameters.

## 2. Materials and Methods

The clinical samples and pertinent clinical information for our study were obtained from July 2013 to April 2014. All of the eligible patients undertaking rotator cuff repair surgeries by the senior author (JYK) and offering valid informed consent form were recruited. These eligible participants were 18–80 years of age and had image findings (MRI or sonography) indicative of a complete or partial tear of the rotator cuff; their discomforts had persisted for at least 3 months in spite of medication and rehab programs. Patients with cancers, chronic renal or liver disease, and non-SS shoulder diseases (previous surgery, fracture, glenohumeral joint osteoarthritis, and instability) were all excluded. The eligible participants were assigned to the patients with and without SS.

SS was defined as 50% decrease of passive ROM for ≥3 months, with normal ROM determined to be 180° abduction, 180° forward flexion, 90° internal rotation, and 90° external rotation. Baseline functional scores and ROM were documented on the day before surgery. These ROM measurements were then summed together to determine the total ROM. Patients were deemed to have SS if total ROM was <270°. We have been employed these criteria in the serial studies published by our team [3,18].

The operation entailed brisement manipulation, anterior acromioplasty, adhesiolysis, limited bursectomy, and rotator cuff repair. After general anesthesia, the patients were placed in a beach chair position. Brisement manipulation following the order of abduction, flexion, external rotation, and internal rotation was performed gently. A saber-shaped incision was performed, and the deltoid muscle was judiciously detached from the acromion and incised downward for 2 cm. One flat elevator was introduced beneath the acromion under arm traction, and a thin osteotome was employed to trim the impinging osteophyte off the acromion. The subacromial fluids (0.5~1 mL) were immediately obtained from stiff and non-stiff shoulders after anterior acromioplasty by sterile syringes. Limited subacromial bursectomy was performed. Extensive adhesiolysis was achieved by excising the contracted coracohumeral ligament and lysing the adhesions around the cuff. Intra-articular irrigation distension of the glenohumeral joint was achieved via the ruptured tendon or via the cleft on the rotator interval. The ruptured cuff was repaired with No. 2 braided polyester sutures, tendon-to-bone suture, or Revo screw (Conmed Linvatec, Edison, NJ, USA) application. The detached deltoid was sutured back onto the acromion before skin wound closure. A triangular bandage was employed for postoperative protection.

The obtained subacromial fluid was centrifuged at 12,000× *g* at 4 °C for 1800 s before storage at −20 °C. The harvested subacromial bursa were immersed in liquid nitrogen and homogenized for immunoblotting. The homogenized subacromial bursa were processed by the Pro-Prep lysis buffer (iNtRON Biotechnology, Sangdaewon-Dong, Korea) and probed by the antibodies against leptin (Thermo Fisher Scientific, Waltham, MA, USA). The blotting intensity was quantified by the AutoMeasure software (Zeiss Inc., Oberkochen, Germany). Immunoreactivity was represented as the proportion of positively probed cells/total cells for 3 histologic sections per patient.

The reverse transcription-polymerase chain reaction (RT-PCR) assay was employed to compare the transcription levels of leptin and the genes of interest in the stiff and nonstiff shoulder retrieved bursa tissues. Tissues were ground under liquid nitrogen free from RNAase, and the total RNA was extracted by an RNA purification kit (RiboPure, Thermo Fisher Scientific). One microgram of extracted RNA was reverse-transcribed into complementary DNA (Step One Real-Time PCR System; Thermo Fisher Scientific) following the manufacturer’s instructions [3]. The threshold cycle (Ct) was determined as the cycle number when the fluorescence signal became identifiable. The 8S ribosomal RNA (18S rRNA) was chosen as the reference gene. The ΔCt was defined by the equation
ΔCt = Ct (18S rRNA) − Ct (target gene rRNA) 

The mRNA level of target genes was defined as 2^ΔCt^.

During cuff-repair surgery, subacromial fluid was extracted for enzyme-linked immunosorbent assay (ELISA). The level of leptin in the subacromial fluid was quantified by the ELISA kits (R&D Systems, Minneapolis, MN, USA) following the manufacturer’s instructions.

In order to investigate the effect of leptin on the synovial fibroblasts obtained from the stiff-shoulder retrieved bursa tissues, about 10^5^ synovial fibroblasts from the stiff-shoulder retrieved-subacromial bursae were treated with graded concentrations of leptin (0 μM, 1 μM, and 3 μM) for 24 h. The transcription expression of interleukin-6, interleukin-10, interleukin-13, and interleukin-1β was quantified by RT-PCR and was repeated thrice.

All of the values are given as the mean ± standard deviation. Between-group differences were assessed for significance using the ANOVA test, and Tukey HSD was employed for post-hoc analysis. The statistical difference was considered to be significant if the *p* value was <0.05. The priori power calculation (G*Power 3.1.9.2 software: http://www.gpower.hhu.de/en.html, accessed on 8 October 2022) used a 2-tailed Wilcoxon signed-rank test to calculate the sample size of at least 26 for each group (calculated effect size: 0.8; α level: 0.05; power: 0.8; and allocation ratio: 1) (access date: 1 July 2022) [19].

## 3. Results

There were 62 patients with rotator cuff lesions submitting valid informed consent for the study between July 2013 and April 2014: 28 patients (23 female, 5 male) with SS and 34 patients (23 female, 11 male) without SS. The constitution of gender, side, age, body-mass index, and presence of diabetes was comparable between the patients with and without SS. The patients with SS had a significantly lower constant score and a significantly lower total range of motion, flexion, abduction, and internal and external rotation (Table 1).

The immunoblotting of the subacromial bursa obtained from the patients with and without SS demonstrated that the subacromial bursa obtained from the patients without SS were more heavily stained with leptin than from SS patients (*p* < 0.001) (Figure 1). RT-qPCR showed that the leptin mRNA levels were lower in the subacromial bursa tissues obtained from the patients with SS than from the patients without (Figure 2).

Subacromial fluid was also obtained during rotator cuff repair surgery for enzyme-linked immunosorbent assay (ELISA). We showed that the leptin protein levels in the subacromial fluid obtained from stiff shoulders were markedly lower than those from the nonstiff shoulders (*p* = 0.011) (Figure 3).

We also tried to demonstrate the correlation between the leptin level in the subacromial fluid and the clinical parameters (constant score, total range of motion, flexion, abduction, external rotation, and internal rotation) (Table 2). We showed that leptin level in the subacromial fluid was positively associated with the Constant score (Figure 4), total range of motion (Figure 5), flexion (Figure 6), abduction (Figure 7), and external rotation (Figure 8). However, no correlation between leptin level and internal ration could be observed (Figure 9).

In order to elucidate the impact of leptin on the synovial fibroblasts obtained from the stiff-shoulder retrieved bursa tissues, the synovial fibroblasts derived from stiff shoulder-retrieved subacromial bursa were treated by 0, 1, and 3 μM leptin, respectively. The transcription levels of inflammatory cytokines were quantified by RT-qPCR. Under RT-qPCR analysis, leptin was shown to dose-dependently decrease the transcription of IL-6 (Figure 10), IL-10 (Figure 11), and IL-13 (Figure 12), but without an impact on IL-1β (Figure 13) and IL-4 (Figure 14) (*p* < 0.001, *p* = 0.001, *p* = 0.001, *p* = 0.137, and *p* = 0.883 by ANOVA test, respectively).

## 4. Discussion

Although the pathogenesis of SS is not fully delineated, it appears that it is fibroblastic and inflammatory in nature [20].

As for the fibroblastic aspect, the samples obtained from cadavers and from arthroscopic surgeries demonstrated synovial hyperplasia with increased vascularity in the initial stage of SS and early-stage fibrosis in the rotator cuff interval, the base of the coracoid process, and the subscapular recess [21]. The thickening of the coracohumeral ligament is one of the specific presentations of SS and the main limitation for external rotation, although it also contributes to limited internal rotation [22]. In advanced SS, the contraction and thickening of the glenohumeral capsule restrain the range of movement in all directions [23]. The findings mentioned above underscore the fibroblastic component of SS.

As for the inflammatory aspect, synovial biopsies from SS patients have demonstrated the expression of cytokines (e.g., IL-1β, IL-6, IL-8, matrix metalloproteinases, and tumor necrosis factor-alpha) and the chronic presence of immune cells (B and T lymphocytes, mast cells, and macrophages) in the stiff shoulders [24,25]. The repertoire of the inflammatory cells and the factors mentioned above underscore the inflammatory component of SS.

Leptin has been shown to modulate the fibroblastic and inflammatory processes of hepatic cholestasis, hepatic fibrosis, and chronic allergic airway disease [26,27,28]. Type II diabetic patients have been shown to have a lower serum leptin level and are prone to suffer from SS [14,15,29]. The recognized role of leptin in mediating the fibroblastic and inflammatory processes as well as the known diabetes-leptin/diabetes-SS linkages inspired us to investigate the role of leptin in the pathogenesis of SS.

Actually, in addition to knee osteoarthritis mentioned above, leptin has already been implicated in various shoulder disorders. Greater leptin and adiponectin levels in the synovial fluid of the osteoarthritic shoulder joint were found to be associated with greater pain among shoulder osteoarthritic patients awaiting arthroplasty surgery [30]. Fasting serum leptin levels were shown to be correlated with the pace of recovery from upper extremity soft tissue disorders (UESTDs) [31].

The correlation between leptin and diabetes has been frequently discussed. In one middle-east study, fasting-lipid-profile, hemoglobin A1c, serum leptin, insulin, and glucose levels were measured among type 2 diabetic patients. The serum leptin level in type 2 diabetic patients (19.32 ± 11.43 ng/mL) was significantly lower than that in non-diabetic subjects (32.16 ± 11.02 ng/mL). Leptin related to homeostasis model assessment for insulin resistance (HOMA-IR) (r = 0.422, *p* = 0.006) was observed among type II subjects [15]. SS is also commonly seen in patients with diabetes. Potential mechanisms include impaired microcirculation and non-enzymatic glycosylation processes around shoulder-joint tissues and the synovium [32].

Despite the known role of leptin in coordinating the fibroblastic and inflammatory processes, the major mechanism underlying SS, as well as the diabetes-leptin/diabetes-SS linkages, the role of leptin in the pathogenesis of SS is not clear at present. According to our study, leptin expression levels are lower among SS patients, and the leptin level in the subacromial fluid was positively associated with constant score, total range of motion, flexion, abduction, and external rotation. These findings seemingly contradict the impression of the pro-inflammatory nature of leptin. We hypothesize that the pro- and anti-inflammatory nature of leptin could vary depending on the tissue leptin acts on. In our study, we also showed that leptin could dose-dependently decrease the transcription of IL-6, IL-10, and IL-13, but without an impact on IL-1β and IL-4.

The role of IL-6 in the pathogenesis of SS has been discussed in previous studies. Moeed et al. collected shoulder capsule tissues from 10 SS patients and 10 patients undergoing shoulder stabilization surgery. Fibroblasts cultured from SS capsule tissues produced more IL-6 in comparison to control fibroblasts, and exposing control fibroblasts to IL-1β markedly increased stromal activation marker transcript and protein expression [20]. Yang et al. found that synovial fibroblasts from stiff shoulders synthesized more from the extracellular matrix than that from non-stiff shoulders. RNA-sequencing and bioinformatic analysis indicate that IL-6 is the most relevant gene in promoting the fibrosis of synovial fibroblasts. The expression levels of IL-6 in synovial fibroblasts from stiff shoulders and IL-6 in culture supernatant were both significantly increased. The authors thus concluded that IL-6 is upregulated in synovial fibroblasts from stiff shoulders and promoted the fibrosis processes [33]. One prospective comparative study recruited patients undergoing arthroscopic treatment for SS and control patients being treated for subacromial bursitis. Synovial biopsies were taken from all subjects. Cytogenetic analysis revealed that the expression of IL-6 was elevated in SS cases as compared to controls [24].

Although the roles in the disease processes of SS are less directly delineated at present, both IL-10 and IL-13 have been implicated in the fibroblastic processes of other organs. IL-10 is a fibrosis-related inflammatory mediator involved in the disease progression of pulmonary fibrosis, renal fibrosis, interstitial fibrosis, cardiac fibrosis, and liver fibrosis [34]. The serum levels of IL-10 were significantly higher among patients with rotator cuff tear with sleep disturbance compared with those of the control group (*p* = 0.05) [35]. This study nicely demonstrated the association between IL-10 and pain perception, and our study showed that leptin could lead to decreased IL-10 expression, complying with the observation that leptin expression was decreased among SS patients. IL-13 has also been shown to mediate the processes of tissue fibrosis via the transforming growth factor-β-dependent and -independent pathways [36,37].

In our study, we found that leptin could dose-dependently inhibit the transcription of IL-6, IL-10, and IL-13, three fibroblastic-promoting genes. The decreased leptin expression in stiff shoulders complies with the decreased transcription of fibroblastic-promoting genes.

One major limitation of our study entails that the scarce volume of subacromial fluids (0.5 mL and 1 mL, at most, from stiff and non-stiff shoulders, respectively [3]) did not allow for comparison of the concentration of IL-6, IL-10, and IL-13, recognized leptin-suppressed cytokines, between SS and non-SS groups by ELISA assay. The repetitive aspiration of subacromial fluids from SS patients may be warranted to attain an adequate volume for analysis in future studies.

## 5. Conclusions

Our study demonstrated that the transcription and protein-synthesis of leptin were significantly lower in the subacromial bursa retrieved from stiff shoulders than from non-stiff shoulders. The protein concentration of leptin in the subacromial fluid was lower in the SS group than in the control group and was positively associated with the Constant score, total range of motion, flexion, abduction, and external rotation of the affected shoulder. Leptin dose-dependently suppressed the transcription of IL-6, IL-10, and IL-13, but without an impact on IL-1β and IL-4 (Figure 15). These results supplemented our knowledge on the role of leptin in orchestrating the disease processes of SS.

## Figures and Tables

**Figure 1 life-12-01588-f001:**
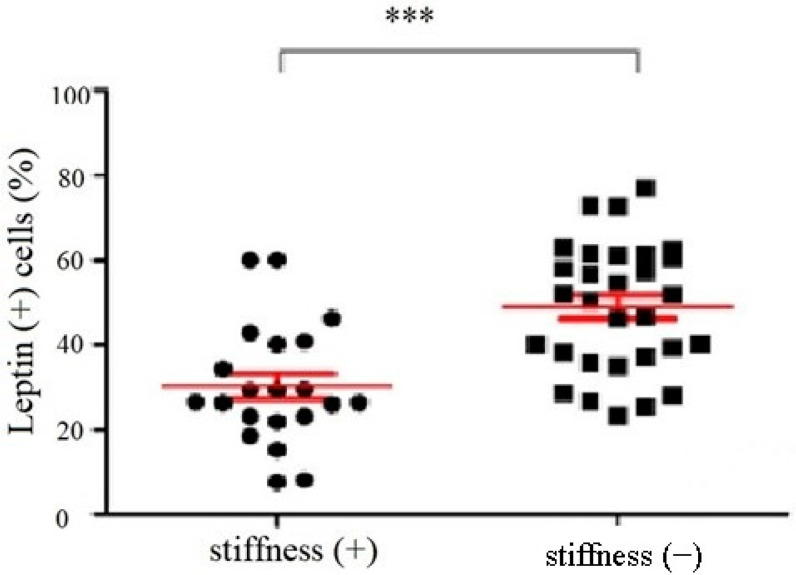
The percentage of leptin positive cells in the subacromial bursa obtained from the patients with and without SS under immunoblotting (***: *p* < 0.001).

**Figure 2 life-12-01588-f002:**
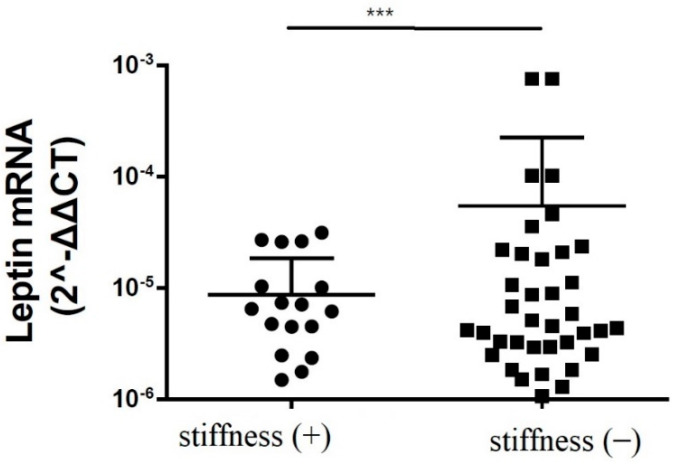
The mRNA expression levels of leptin in the subacromial bursa obtained from the patients with and without SS (***: *p* < 0.001).

**Figure 3 life-12-01588-f003:**
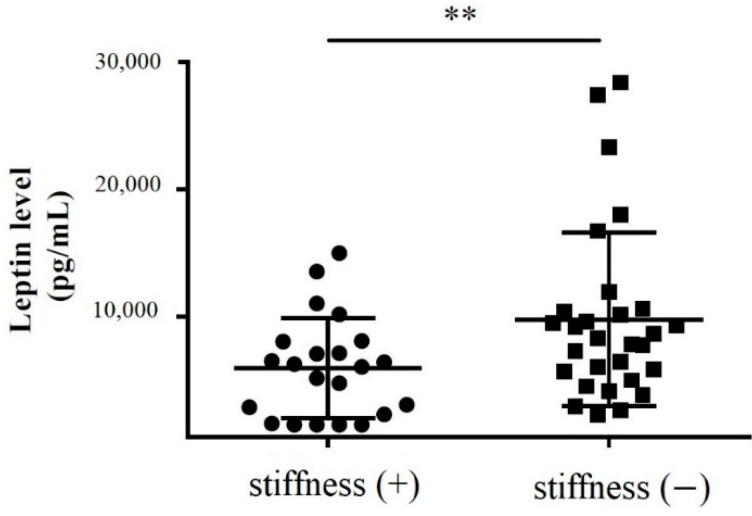
The concentration of leptin in the subacromial fluid obtained from the patients with and without SS (**: 0.001 < *p* < 0.010).

**Figure 4 life-12-01588-f004:**
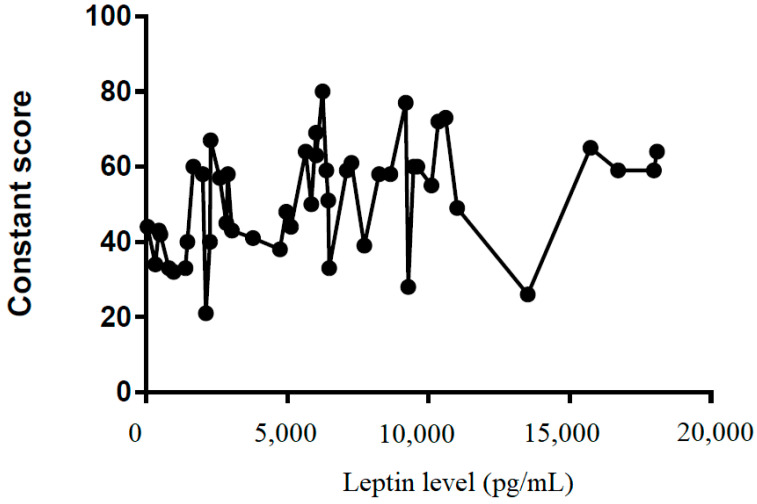
The correlation between Constant score and leptin level in subacromial fluid (Pearson R: 0.39 (0.12~0.61), R^2^ = 0.15, and *p* = 0.006).

**Figure 5 life-12-01588-f005:**
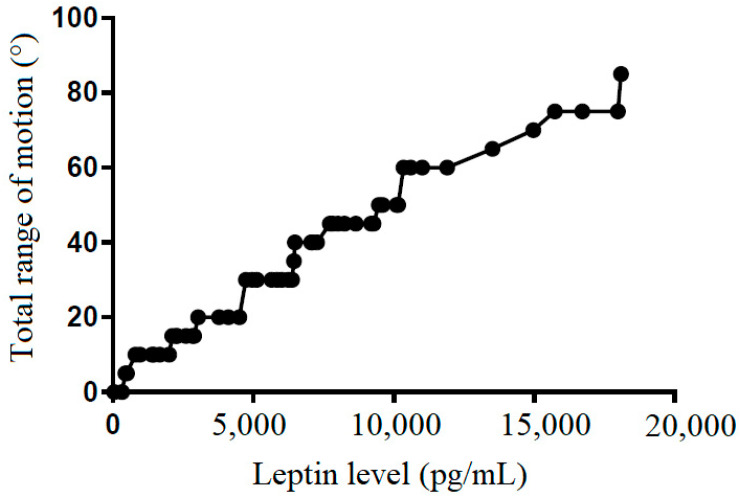
The correlation between total range of motion and leptin level in subacromial fluid (Pearson R: 0.50 (0.52~0.84), R^2^ = 0.53, and *p* < 0.001).

**Figure 6 life-12-01588-f006:**
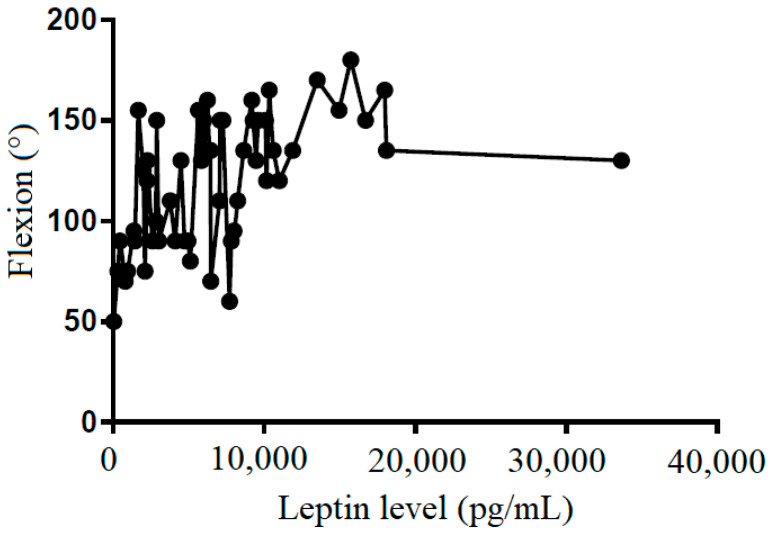
The correlation between flexion and leptin level in subacromial fluid (Pearson R: 0.50 (0.28~0.68), R^2^ = 0.25, and *p* < 0.001).

**Figure 7 life-12-01588-f007:**
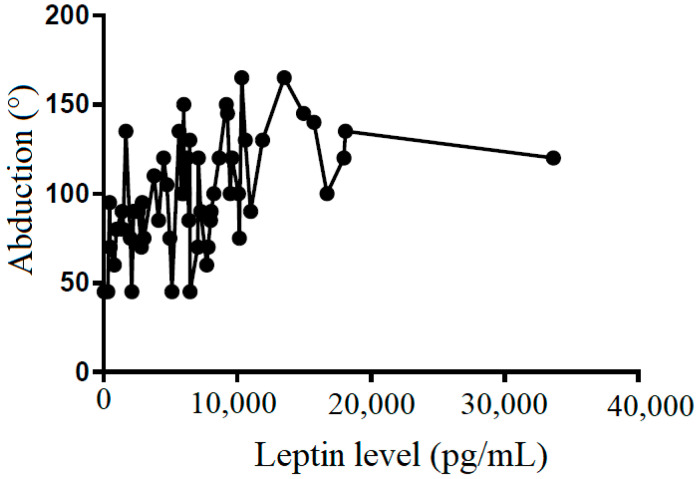
The correlation between abduction and leptin level in subacromial fluid (Pearson R: 0.49 (0.26~0.66), R^2^ = 0.24, and *p* < 0.001).

**Figure 8 life-12-01588-f008:**
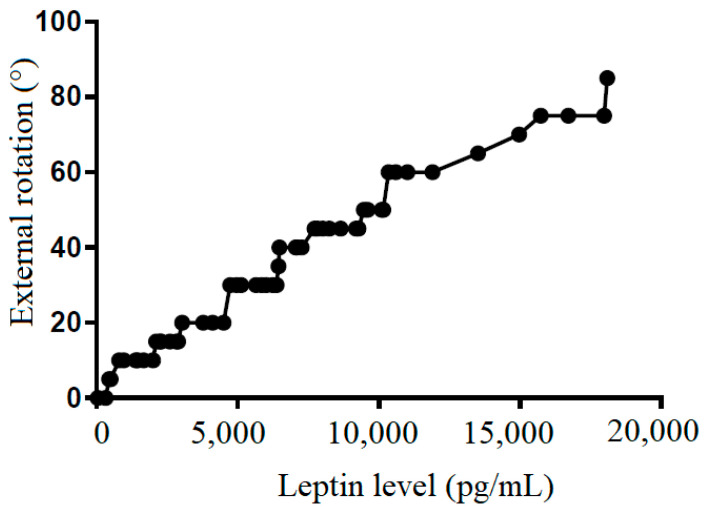
The correlation between external rotation and leptin level in subacromial fluid (Pearson R: 0.99 (0.98~0.99), R^2^ = 0.97, and *p* < 0.001).

**Figure 9 life-12-01588-f009:**
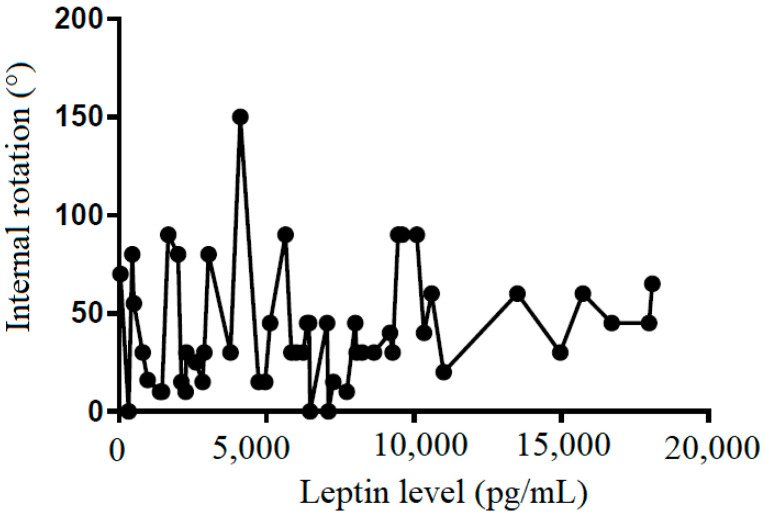
The correlation between internal rotation and leptin level in subacromial fluid (Pearson R: 0.12 (−0.15~0.38), R^2^ = 0.02, and *p* = 0.381).

**Figure 10 life-12-01588-f010:**
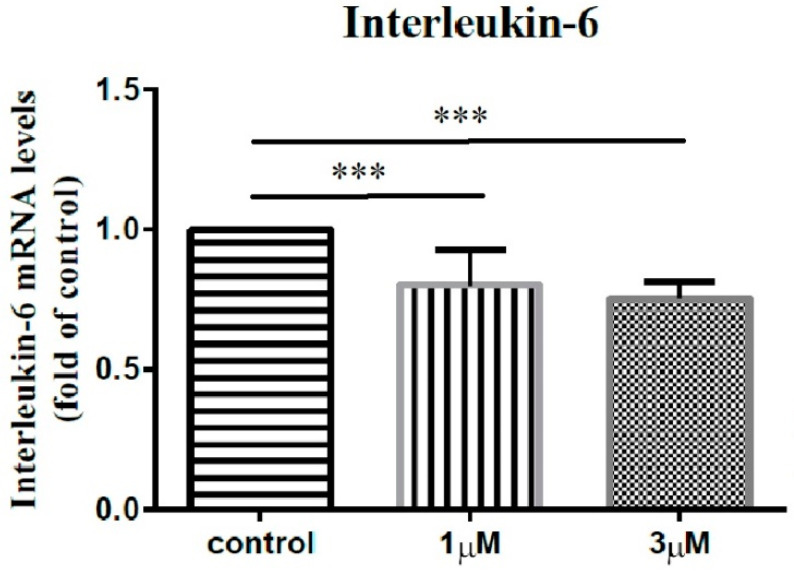
The mRNA expression levels of interleukin-6 of the subacromial bursa cells treated with different concentrations of leptin. (***: *p* < 0.001 by post-hoc analysis).

**Figure 11 life-12-01588-f011:**
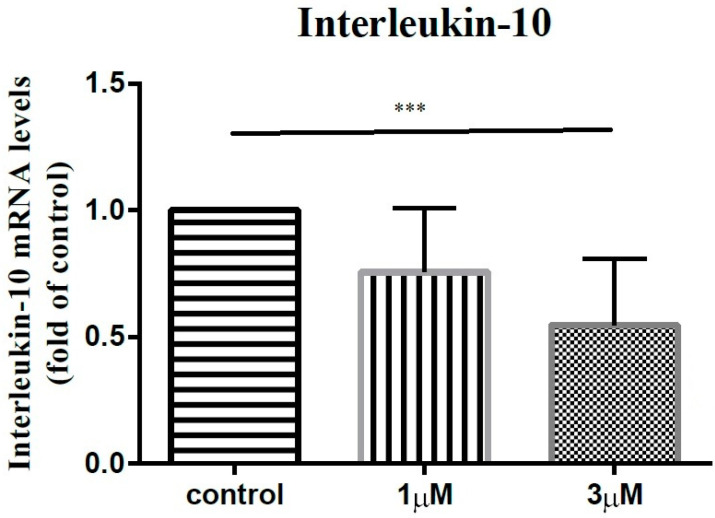
The mRNA expression levels of interleukin-10 of the subacromial bursa cells treated with different concentrations of leptin. (***: *p* < 0.001 by post-hoc analysis).

**Figure 12 life-12-01588-f012:**
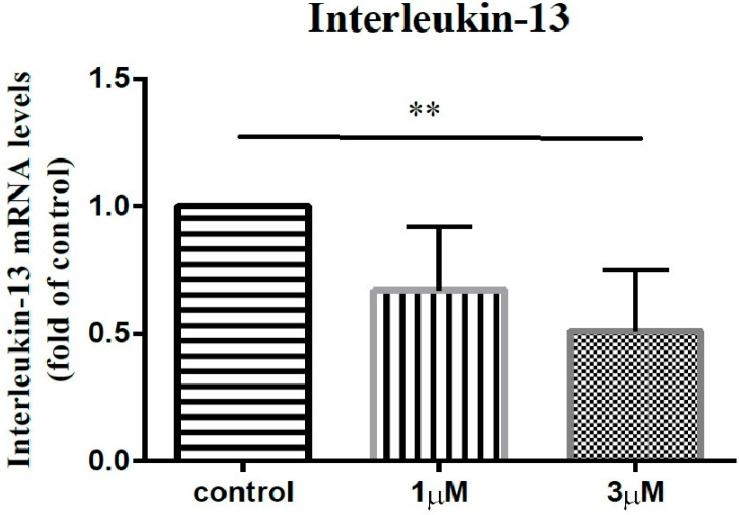
The mRNA expression levels of interleukin-13 of the subacromial bursa cells treated with different concentrations of leptin. (**: 0.010 > *p* > 0.001 by post-hoc analysis).

**Figure 13 life-12-01588-f013:**
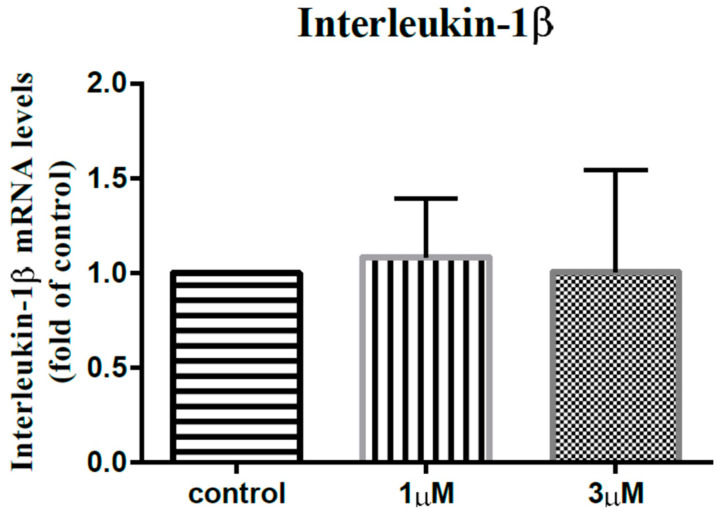
The mRNA expression levels of interleukin-1β of the subacromial bursa cells treated with different concentrations of leptin.

**Figure 14 life-12-01588-f014:**
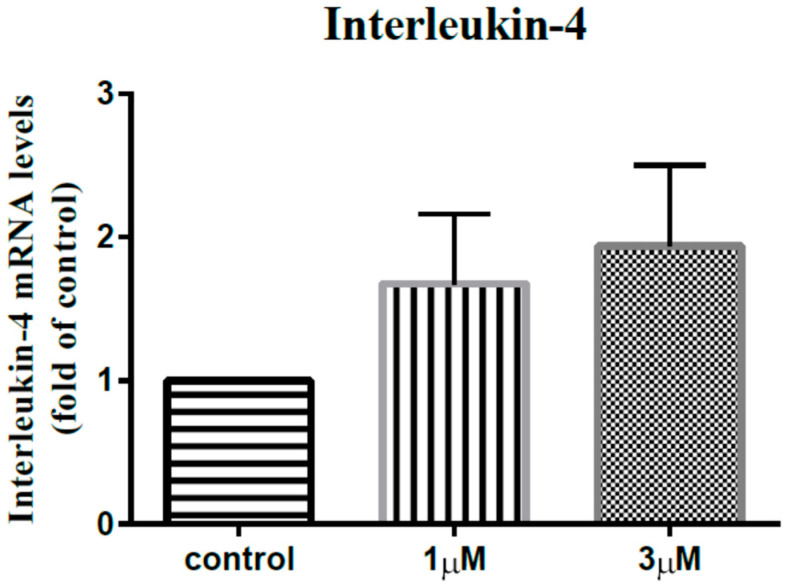
The mRNA expression levels of interleukin-4 of the subacromial bursa cells treated with different concentrations of leptin.

**Figure 15 life-12-01588-f015:**
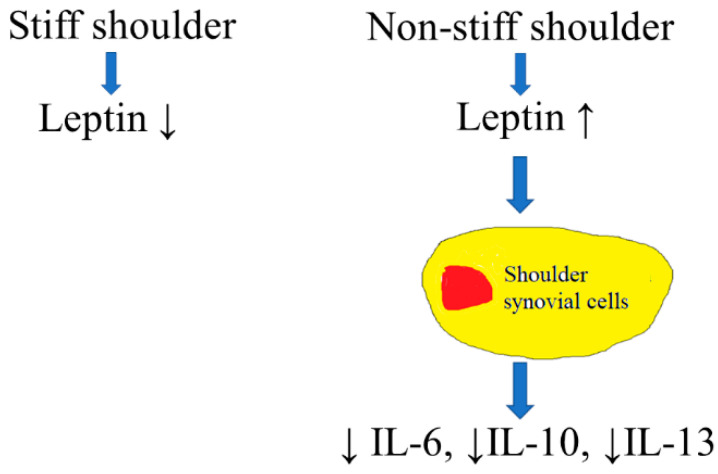
An illustration of the role of leptin in the pathogenesis of SS. Leptin would lead to suppressed expression of IL-6, IL-10, and IL-13 in synovial cells.

**Table 1 life-12-01588-t001:** The demographic details for the patients with and without SS.

	Stiff	Non-Stiff	*p*-Value
Gender			0.194
Male	5	11	
Female	23	23	
Side			0.175
Right	16	25	
Left	12	9	
Age (years)	61.71	62.74	0.332
Body mass index (kg/m^2^)	25.62	25.8	0.902
Diabetes			0.933
No	17	21	
Yes	11	13	
Constant score	40.14	60.35	<0.001
Total range of motion (°)	277.84	354.81	<0.001
Flexion (°)	91.92	142.67	<0.001
Abduction (°)	78.08	116.0	<0.001
External rotation (°)	23.08	44.14	<0.001
Internal rotation (°)	35.64	50	<0.001

**Table 2 life-12-01588-t002:** The correlation between leptin level in the subacromial fluid and the clinical parameters.

	CS	Total ROM	Flexion	Abduction	ER	IR
Pearson R (95% CI)	0.39 (0.12~0.61)	0.73 (0.52~0.84)	0.50 (0.28~0.68)	0.49 (0.26~0.66)	0.99 (0.98~0.99)	0.12 (−0.15~0.38)
R^2^	0.15	0.53	0.25	0.24	0.97	0.02
*p* value	<0.001	<0.001	<0.001	0.001	<0.001	0.380

CS: constant score; ROM: range of motion; ER: external rotation; IR: internal rotation; and 95% CI: 95% confidence interval.

## Data Availability

The data presented in this study are available on request from the corresponding author.

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
