# Peer review of "Decreased Expression of Leptin among Patients with Shoulder Stiffness"

_life, 2022, doi:10.3390/life12101588_

Round 1

Reviewer 1 Report

The work seems interesting I have the following comments:

1. Introduction

1.     Please add, possible physiotherapy treatments for SS.

2.     Statistics on SS epidemiology and information on treatment costs.

3.     Please clearly formulate the objective and add the research hypothesis.

2. Materials and Methods

4.     In the statistics, please add the effect size.

3. Results

5.     Table 1: It is incomprehensible to me to analyze ''Gender (male/female)''. Is this an analysis in terms of the number of people in the control and study groups? If so, please add an analysis comparing only the number of males and an analysis comparing only the number of females between groups.

6.     Why did the authors decide on such a number of people? Was it dictated by the analysis of the multiplicity of the group? Please include this information in the text.

7.     The authors variably give values statistically once to the third decimal place and once to the fourth. Please standardize the notation.

5. Conclusions

8.     The description of the authors' conclusions is too much repetition of the results suggests rephrasing the entire conclusions.

Author Response

Please see the attached PDF file. 

Reviewer 2 Report

jIh-Yang Ko et al. have investigated into expression of leptin among patients with shoulder stiffness vs no shoulder stiffness and studied the effects of it on inflammatory cytokines and interleukins.

The authors concluded that patients with shoulder stiffness has significantly reduced leptin levels. They also showed the leptin can reduce the levels of inflammatory interleukins such as interleukin-6, 10 and 13.

The manuscript can not accepted in its current form. The manuscript requires following changes:

1) In the simple summary part, line 16 and 17, the authors claim that "Leptin could enhance the transcription of interleukin-6, interleukin-10, interleukin-13 and interleukin-1B." First change is required here is that Leptin has shown to reduce (Figure-10, 11, 12) the levels of these interleukin levels, thus The sentence needs to be modified supporting the data here. Also, there is no change in interleukin-1B (Figure-13), So, interleukin-1B is needed to be removed from the sentence.

2) Page-4, Line 147 to 149, says that patients with SS were more heavily stained with leptin than from patients without. Figure-1 does not support this claim. We observe more laptin (+) cells in Stiffness (-) sample then stiffness (+). Also, figure-2 shoes that leptin mRNA levels are higher as well.

3) The authors should also add data for interleukins between Stiffness (+) vs stiffness (-). That should strengthen the conclusions in the study.

4) The authors should add the summary image that should summarize the findings. 

Round 2

Reviewer 2 Report

The authors have addressed all the concerns raised in the earlier review. Thus, the updated manuscript can now be accepted in its current form.